# HPV DNA Integration at Actionable Cancer-Related Genes Loci in HPV-Associated Carcinomas

**DOI:** 10.3390/cancers16081584

**Published:** 2024-04-20

**Authors:** Xavier Sastre-Garau, Lilia Estrada-Virrueta, François Radvanyi

**Affiliations:** 1Department of Pathology, Centre Hospitalier Intercommunal de Créteil, 40, Avenue de Verdun, 94010 Créteil, France; 2Institut Curie, PSL Research University, CNRS, UMR 144, 75005 Paris, France; lilia.estrada-virrueta@curie.fr (L.E.-V.); francois.radvanyi@curie.fr (F.R.)

**Keywords:** HPV, viral oncogenesis, HPV DNA integration, therapeutic target

## Abstract

**Simple Summary:**

In many tumor types, the development of personalized treatments has been allowed by the optimal molecular characterization of tumor cell genome alterations. In most carcinomas associated with human papillomaviruses (HPV), the integration of part of the viral genome into the tumor cell genome may lead to alterations of cancer-related genes located at the integration locus. In order to assess the overall frequency of such an event, we analyzed a large series of cases for which HPV integration sites had been determined. We found that 40% of the genes located at highly (≥3) recurrent HPV insertions corresponded to cancer-related genes. Moreover, about one-third of the genes targeted by HPV insertions correspond to actionable targets. These observations should lead to a more systematic analysis of HPV DNA integration patterns in HPV-associated carcinomas in order to identify highly specific tumor markers and develop personalized anti-tumor therapies.

**Abstract:**

In HPV-associated carcinomas, some examples of cancer-related genes altered by viral insertion and corresponding to potential therapeutic targets have been described, but no quantitative assessment of these events, including poorly recurrent targets, has been reported to date. To document these occurrences, we built and analyzed a database comprised of 1455 cases, including HPV genotypes and tumor localizations. Host DNA sequences targeted by viral integration were classified as “non-recurrent” (one single reported case; 838 loci), “weakly recurrent” (two reported cases; 82 loci), and highly recurrent (≥3 cases; 43 loci). Whereas the overall rate of cancer-related target genes was 3.3% in the Gencode database, this rate increased to 6.5% in “non-recurrent”, 11.4% in “weakly recurrent”, and 40.1% in “highly recurrent” genes targeted by integration (*p* = 4.9 × 10^−4^). This rate was also significantly higher in tumors associated with high-risk HPV16/18/45 than other genotypes. Among the genes targeted by HPV insertion, 30.2% corresponded to direct or indirect druggable targets, a rate rising to 50% in “highly recurrent” targets. Using data from the literature and the DepMap 23Q4 release database, we found that genes targeted by viral insertion could be new candidates potentially involved in HPV-associated oncogenesis. A more systematic characterization of HPV/host fusion DNA sequences in HPV-associated cancers should provide a better knowledge of HPV-driven carcinogenesis and favor the development of personalize patient treatments.

## 1. Introduction

A subset of human papillomaviruses (HPV) genotypes, classified as carcinogenic to humans (group 1) (HPV16, 18), probably carcinogenic (group 2A) (HPV31, 33), and possibly carcinogenic (group 2B) (some types other than HPV16, 18, 31, 33) [1], are able to infect the mucosa of the ano-genital and oro-pharyngeal tracts and express oncogenic properties, among which HPV16 and HPV18 are the most prevalent [2,3]. Despite the development of prophylactic vaccines against potentially oncogenic HPVs, HPV-associated carcinomas remain a health burden worldwide [4,5]. HPVs are associated with most cervical [6], vaginal [7], and anal [8,9] cancers and with 25–35% of head and neck (H&N) carcinomas [4] recently estimated to be more than 50% in Western countries [10]. At the molecular level, the HPV genome consists of a 7.8Kbp double-stranded circular DNA molecule with oncogenic properties largely related to the E6 and E7 genes. These encoded proteins have the ability to inhibit the p53 and pRB host tumor suppressors, respectively, as well as many other cellular protein targets [11]. In preinvasive lesions, the viral genome commonly replicates extra-chromosomally in the cell nucleus, whereas, in most invasive carcinomas, at least a portion of the viral DNA is stably integrated into a unique or within a few sites in the tumor cell genome [12,13]. These observations led to the hypothesis that integration might represent a step in HPV-related oncogenesis [14]. Upon integration, there is often a disruption of the E1/E2 genes, which causes constitutive expression of the E6/E7 viral oncoproteins [15]. Further knowledge of the consequences of HPV integration in the cell genome progressively improved with the development of several technologies allowing both localization of the viral inserts and analysis of target sequences. Early reports based on in situ *hybridization* (*ISH*) on chromosomes obtained from cervical tumor-derived cell lines demonstrated the localization of HPV inserts at common fragile sites [16,17,18,19]. Using cloned cellular sequences adjacent to viral insertions as probes, HPV18 DNA was found 40Kbp upstream of *MYC* in HeLa cells, raising the hypothesis that HPV insertion can act in oncogenesis [20]. The identification of *MYC* as a recurrent integration site for HPV16/18 [21] and the observation of HPV insertions at translocation breakpoints [22] further supported the hypothesis of oncogene cis-activation via the insertion of viral DNA. However, the data remained scarce due to the bias introduced by the laborious cell culture steps and by the limited resolution of ISH.

Studies using PCR-based approaches, allowing the analysis of a large number of cases, confirmed that the HPV integration process was a clonal event, more frequent in invasive than in intraepithelial neoplasia, and could thus act in tumor progression [23,24,25]. A wide dispersion of mapped viral insertions throughout the genome was observed, frequently near fragile sites, but alterations of cellular genes located at the insertion sites were rarely observed [24,25]. Nevertheless, certain studies underlined the non-random distribution of HPV inserts in the genome [26], while others observed that host genome regions with high transcriptional activity were preferentially targeted by HPV insertions [27,28]. It was further shown that HPV integration frequently targeted genes [29] and was associated with structural alterations of the cell genome [30]. Some examples of gene activation [31,32] and gene inactivation [33,34] through HPV insertion were reported.

The era of NGS has profoundly modified the approach by allowing, in the same experiment, the characterization of both the entire viral sequences and the pattern of the viral/host fusion sequences. A series of cervical [35,36], H&N [37,38,39], anal [40], and vulvar [41] cancers could be analyzed using multiplex capture of HPV inserts. Improved accuracy of the rate of HPV integration in cervical tumors was obtained, ranging from 71.4% [35] to 92.2% [42] (mean 387/446 = 86.8%) and up to 100% in HPV18-associated tumors [36,43]. This rate was 60%-77% in H&N [37,38,39,44] and 54.8% in anal cancers [40]. The proportion of multiple viral insertions in individual cases could also be assessed more precisely. In the TCGA analysis [43], one single viral integration site was observed in 64% of the cases, two sites in 25%, and more than two in 11%. These studies also confirmed the integration hot spots previously reported (*MYC*/*PVT1*) and identified new recurrent insertions targeting *TP63*, *RAD51B*, *MACROD2*, *KLF5*/*KLF12*, *PDL1*/*PDL2*/*PLGRKT*, and *TTC6*/*MIPOL1* [35,36,37,38,39].

The co-analysis of genomic and expression patterns has further reinforced the knowledge of the molecular consequences of HPV integration in tumor cells [37,39,42,43,45,46,47]. Fusion HPV genome transcripts included known or predicted genes in 70% of the integration events [43], and overexpression of the genes located near integrated viral sequences was detected in about 50% of the cases [37,45,46,47]. High transcriptional changes, preferentially observed in tumors harboring viral inserts in exonic regions [37], might be related in part to local genomic amplification upon HPV insertion, but cases of overexpression without amplification were also observed [45]. One study found silent and scattered viral inserts in tumors with multiple insertions [42]. These studies also provided some examples of gene disruption via viral insertion within *RAD51B*, implicated in homologous DNA repair mechanisms [39].

Several recent reviews have summarized the accumulated viral integration data in HPV-associated tumors [47,48,49,50,51]. In 2016, the studies performed by Zhang et al. [51] and Bodelon et al. [48] based on the analysis of 499 and 1.500 integration events, respectively, found that HPV DNA preferentially integrated into gene-dense regions, close to enhancers in transcriptionally active regions of the human genome and in intragenic loci. Most of the recurrent target genes in chromosomal hot spots were functionally cancer-related. In 2021, Warburton et al. [50] investigated 1.418 integration breakpoints from cervical and H&N carcinomas and observed multiple clustered HPV integration breakpoints associated with amplified regions of the host genome encompassing gene loci related to cell development and identity. Expression analyses showed that the majority of tumors had only one single transcriptionally active driver integration locus. HPV integration breakpoints were enriched at both *FANCD2*-associated fragile sites and enhancer-rich regions. The authors stressed that the driver of oncogenesis via HPV integrants requires a combination of events dependent on the genetic and/or epigenetic landscape of the flanking sequences.

Altogether, these studies have provided unambiguous data, indicating that HPV DNA insertion could act in oncogenesis via a direct interaction with the cell genome. However, it is not clear whether this model of cancer-related gene alteration via HPV insertion is mainly restricted to a limited number of observations revealed by their recurrency or corresponds to a more general mechanism also commonly implicated at multiple sites with low recurrency levels and may, therefore, constitute a rational basis for new strategies of therapeutic intervention. This led us to constitute and analyze a large database in order to (I) define whether the rate of multiple vs. single integration pattern within the same tumor was dependent on HPV genotypes; (II) analyze whether recurrent HPV insertions at certain host genome targets were related to a specific HPV genotype and/or tumor localization; (III) evaluate the rate of cancer-related genes targeted by viral insertion in recurrent vs. non-recurrent integration spots; and (IV) provide an assessment of insertional targets and/or pathways that might be actionable for personalized therapy development.

## 2. Materials and Methods

Ethical review and approval were waived for this study, which is based on literature data mining and analysis. In order to assess the rate of cancer-related gene alteration via HPV insertion in HPV-associated tumors, we first established a large database of HPV inserts, including tumor localization, tumor type, HPV genotype, and gene(s) targeted by insertion. We then separated the target genes into three groups: a group of “non-recurrent genes” (only one case of HPV insertion reported at this locus), a group of “genes with low recurrency level” (two cases of HPV insertion reported at this locus), and a third group of “genes with high recurrency level” (≥3 cases reported). In a third step, we compared the rate of tumor-related genes at HPV insertion in each of these three categories, respectively, and finally, we looked for the rate of actionable genes among HPV insertion targets in the general population of tumors and in each of the three groups of genes defined according to their recurrency level.

Our database was built using results from the literature published between 1987 and 2023. A total of 58 publications were registered [13,16,17,18,19,20,21,22,24,25,28,29,30,31,32,33,34,35,36,37,38,39,41,42,45,47,52,53,54,55,56,57,58,59,60,61,62,63,64,65,66,67,68,69,70,71,72,73,74,75,76,77,78,79,80,81,82,83]. For each case, we documented the chromosomal locus of HPV insertion, HPV genotype, organ, type of lesion, sample identification, first author, date of publication, review, technique used, target gene, and coordinates of the HPV insertion when available. A list of 1455 integration loci was established. The registration of sample identification permitted us to discard cases reported several times in different publications and analyze cases with multiple integration sites according to their viral and clinical characteristics. From the analysis of the number of sites per tumor, we discarded a tumor-derived cell line that might harbor several sites developed in vitro and 9 cases associated with more than one HPV genotype that could correspond to different lesions in the same tumor diagnostic tissue sample. From the analysis of genes targeted by integration in each tumor, we excluded 34 cases explored by *ISH* only and 69 cases for which the sample identification was not specified. A total of 1031 cases were retained for viro-clinical analyses.

To analyze if the HPV-inserted genes were significantly enriched in cancer-related genes or in therapy targets, we compared them to a reference panel of genes without HPV inserts composed of protein-coding genes from Gencode [84] v42hg19 annotation. For the analysis, we kept only protein-coding genes with HPV insertions that were present on Gencode, removing 64 genes that were not found. Certain gene names were changed to be compatible with the Gencode v42 version, and 727 genes (1012 cases) were retained in our analysis.

The genes were classified according to their role in cancer, their role in cell proliferation in vitro using the DepMap score [85], and whether they are actionable or not. The cancer pathway for each gene was estimated according to the data from the OncoKB database [86,87]. The genes were considered to be actionable according to the Therapeutic Target Database [88] (TTD) and the Drug-Gene Interaction Database [89]. Figures and statistical tests were generated using R version 4.3.2. For the statistical analysis, we performed Fisher’s exact test, comparing the genes according to their recurrency status (0, 1, 2 ≥ 3). The contingency matrix was constructed considering the recurrency status of each gene vs. its role in cancer and vs. whether it is actionable or not. The insertion loci were also classified according to the HPV genotype (no HPV insertion; HPVs other than HPV16,18,45 (referred to as “intermediate risk HPV genotypes”); high-risk HPV16 and high-risk HPV18/45), and the contingency matrix was computed considering the role in cancer of the inserted gene. In each case, *p*-values were adjusted using the Holm–Bonferroni correction. For the highly recurrent genes with an unknown role in cancer, we used the DepMap CRISPR gene effect score to identify new candidate genes to be involved in cervical or H&N cancers. We excluded the genes with a very low expression value (TPM < 1) in the respective cell lines. A gene was considered to be a potential cancer gene according to the CRISPR gene effect score if more than 30% of cervical and H&N cell lines had a score below −0.3 for this gene (for the candidate oncogene) and above 0.3 (for the candidate tumor suppressor genes). Flowchart analysis in Figure 1.

## 3. Results

### 3.1. Viral Analysis According to Tumor Localization

The 1031 tumor cases analyzed were developed in the cervix (813; 78.9%), head and neck (H&N) (164; 15,9%), anus (32; 3.1%), vulva (16; 1.5%), vagina (4; 0.4%), and penis (2; 0.2%). Most cases (970; 94%) corresponded to invasive carcinoma or carcinoma-derived cell lines (6; 0.6%). Others were high-grade intraepithelial neoplasia developed in the cervix (38), vulva (8), or vagina (2). Two low-grade intraepithelial lesions and one laryngeal papillomatosis were also recorded (four cervical lesions of undetermined histology) (Appendix A).

Virology data were available in 1.028 cases. The most frequent genotypes were HPV16 (690, 67.1%), HPV18 (184; 17.9%), HPV45 (48; 4.7%), HPV33 (30; 2.9%), HPV58 (15; 1.6%), and HPV31 (15; 1.5%). Other genotypes detected in less than 1% were HPV35 (9), HPV73 (8), HPV68 (5), HPV51 (4), HPV52 (4), HPV56 (4), HPV59 (4), HPV26 (1), HPV34 (1), HPV39 (1), HPV6 (1), HPV11 (1), HPV67 (1), HPV70 (1), and HPV82 (1) (undetermined HPV type in three cases). The comparison between HPV genotypes and tumor localizations showed that HPV18/45 were largely more prevalent in cervical than in H&N tumors (28% vs. 1%). HPV16 was highly predominant in H&N tumors (86%), together with the non-HPV16/18 types (13%), among which HPV33 and 35 were the most frequent (6.7% and 4.9%, respectively) (Figure 1). Other tumor localizations were almost exclusively (96%) associated with HPV16 (Appendix A).

### 3.2. In Individual Tumors, the Development of Multiple HPV DNA Inserts Is Partly Related to the Viral Genotype

The identification and HPV genotyping of samples could be unambiguously specified in 951 cases, allowing the determination of the number of viral inserts in each case. There was a single viral insert per tumor in 754 (78.5%) cases, 2 integration sites in 124 (13.7%) tumors, 3 sites in 33 (3.6%), 4 sites in 16 (1.6%), 5 sites in 11 (1.1%), 6 sites in 3 (0.3%), 7 sites in 4 (0.4%), 9 sites in 2 (0.2%), and 10, 11, 12, and 14 sites in one tumor each (0.1%) (Appendix A). We compared the number of viral inserts per case and the viral genotype. This analysis showed that HPV18 and HPV45 were less frequently associated with multiple insertion sites (8.5% and 0.6%, respectively) than HPV16 (25.0%) or the other HPV types (19.8%) (Figure 2) (*p* = 0.003). For instance, only 3 of the 73 cases with ≥3 insertion sites were associated with HPV18/45.

### 3.3. HPV DNA Preferentially Integrates near or into Genes and Recurrent Targets Are Identified

A total of 1455 insertions were recorded, from lesions developed in the cervix (1145 insertions), H&N (233), anus (53), vulva (18), vagina (4), and penis (2). In 1.186 of these cases (81.5%), a target sequence was identified at the insertion locus. Among these, 72 did not contain known or expressed sequences (Alu, pseudogenes, or sequences of unknown functional significance). Sequences with potential functional relevance were observed in 1104/1455 (75.8%) of the cases. Among these, 125 genes were recurrently targeted by viral insertion. The most frequent were *TP63* (22 cases), *KLF5/12* (21 cases), *MACROD2* (15), MYC (13), VMP1 (12), RAD51B (12), CEACAM (10), PVT1 (8), CD174 (8), FHIT (7), NFIX (7), *C9orf3* (7), *MIPOL1/TTC*, *LRP1B*, *PLGRKT*, *ACTL7B*, *ERBB2* (6 cases each), *CASC21/CASC8*, *TUBD1*, and *BCL11B* (5 cases each) (Figure 3). In addition, 23 genes were targeted in three or four cases each and 82 genes in two cases (Figure 3 and Appendix A) (all data on Appendix A).

The comparison between recurrent HPV insertional targets and tumor localizations showed that all of the insertions at *MYC* (13), *FHIT* (7), and *VMP1* (12) were found in genital tumors and were preferentially associated with HPV18/45, whereas the insertions on *ACTL7B* (6/6), *CD274* (7/8), and *PLGRKT* (4/6) were mainly found in HNCC cases (one *CD274* insertion in anal cancer) and preferentially associated with HPV16. Finally, *NFIX* was the preferential target for HPV16 in anal carcinoma (5/7). Comparing genes at viral integration and HPV genotypes, we found a limited number of hot spots that were preferentially targeted by specific HPV genotypes: *VMP1*, *MYC*, and *LRP1B* were preferential targets for HPV18/45, *PVT1* for HPV16, and *PLGRKT* for HPV33. Other targets (*TP63*, *KLF5/12*…) showed a balanced repartition between the different viral genotypes and tumor localizations (Appendix A).

### 3.4. The Proportion of Target Genes Implicated in Oncogenesis Increases Proportionally with Increasing HPV Insertion Recurrency Levels and with the Oncogenicity of the HPV Genotypes

In order to further assess the targeting frequency of cellular cancer-related genes via HPV DNA insertion, we analyzed separately the proportion of this event in groups of non-recurrent vs. recurrent genes for HPV integration. Of the 792 target genes identified in our series, 727 matched with genes from Gencode. Using the OncoKB Cancer Gene List, these 727 genes were classified into four functional categories: genes with pro-tumorigenic or oncogenic (OG) or with tumor suppressive (TSG) effects, genes expressing both OG and TSG (OG/TSG) effects, and genes with no recognized effect in tumorigenesis (N). We compared the proportion of these functional categories in a panel of 19,402 genes with that observed among genes found targeted in one single tumor case (604 genes), in two cases (79 genes), or in more than two cases (44 genes). The proportion of cancer-related genes was 3.3% in the reference panel of genes. It was 6.5% among the group of genes with a single integration event, 11.4% among genes reported twice as HPV insertion targets, and 40.9% among genes reported more than two times as viral insertion targets (Figure 4A) (*p* = 4.9 × 10^−4^).

We next examined whether the proportion of cancer-related genes targeted by viral integration might be influenced by the genotype of the HPV inserts. For this analysis, we focused on cervical cancers in which the viral heterogeneity was the highest. Viral genotypes were separated into three categories: high-risk HPV16 (514 cases), high-risk HPV18/45 (167 cases), and intermediate-risk non-HPV16/18/45 (82 cases). As compared with cases in the reference data set, the rate of cancer-related genes targeted by integration was higher in tumors associated with intermediate-risk HPVs and significantly increased in tumors with high-risk HPV genotypes that were highest in the group of HPV18/45-associated carcinomas (Figure 4B).

Taken together, the observations that the rate of cancer-related genes increases proportionally to both HPV insertion recurrency level and viral genotype oncogenicity constitute a strong argument for a direct role of viral insertion as a common mechanism in HPV-associated oncogenesis and should lead to the identification of new genes also implied in tumor processes that may constitute attractive therapeutic targets.

### 3.5. The Proportion of Genes Targeted through HPV Inserts and Corresponding to Therapeutic Targets Increases According to Viral Integration Recurrency

We used the DGIdb database to look for the proportion of genes targeted through viral integration that correspond to therapeutic onco-targets or are implied in targetable molecular pathways. About 19% of the 19,402 genes in the database were recorded as direct or indirect therapeutic targets. This proportion was 27.6% among the genes reported as HPV insertion targets in a single case, 40.5% among the genes reported as HPV targets two times, and 50% in the genes reported as targets more than two times (Figure 5) (overall rate 30.2%). The proportion was the same for cervical or H&N tumors. Since several HPV integration sites may be observed in the same tumor, we assessed the proportion of patients for whom a potential therapeutic target could be identified by an HPV insertion pattern as about 26%.

### 3.6. The Localization of the Viral Inserts at the Molecular Level Regarding the Structure of the Targeted Gene Does Not Differ Significantly between Pro-Tumorigenic and Tumor Suppressive Genes

In order to determine whether the integration pattern differed between pro-tumorigenic and tumor suppressor genes, we compared the position of the viral inserts in a few bona fide examples of genes in these two categories. Depmap plots were used to verify the functional properties of these genes in cervical cancer- and H&N tumor-derived cells. Most of the viral inserts were found located in introns or upstream of the genes, regardless of their respective pro- or anti-tumorigenic effect (Figure 6). In particular, no preferential occurrence of viral DNA integration within gene exons, suggestive of disruptive gene alteration, was observed in tumor suppressors.

### 3.7. Identification of New Genes Involved in HPV-Associated Cancers

A total of 78% of the genes recurrently targeted by viral insertion were not classified as oncogene or tumor suppressor genes in OncoKB. At least three explanations could account for this observation: (I) genes not involved in oncogenesis may be targeted by viral insertion regarding their proximity to fragile sites; (II) genes not classified as oncogene or tumor suppressor may nevertheless, upon activation or inactivation, provide an advantage in vivo to cancer cells; for instance, overexpression of *FOXA1* associated with viral insertion (three cases) may induce tumor cells protection from the host immune system; (III) oncogene or tumor suppressor in HPV-associated cancer may not be yet identified as such in OncoKB. To identify the genes belonging to this latter category, we used two strategies: a literature search and the analysis of gene effects using a large CRISPR screen including 17 cervical and 56 H&N cancer cell lines (https://depmap.org/portal/ accessed on 6 December 2023). Among the 43 genes corresponding to hot spots (≥3) targets for HPV insertion, 26 were not referred as cancer-related in OncoKB. The literature search and/or the CRISPR screen allowed us to further classify some of these genes as oncogenes (*AKR1C3*, *TUBD1*, and *VMP1*), tumor suppressors (*LEPREL1*), or genes with dual pro- and anti-tumor effects (*FAM110B*, *KLF12*, and *MAGI2*), while others were found to act in the tumor process as transcription factors (*NAALADL2*, *NFIA*, *NFIX*, and *NR4A2*), cell adhesion (*CEACAM5/6*), cell proliferation (*TPRG1*), matrix modeling molecules (*PLGRKT*), or immunoevasion (*MAPK10* and *CXCL8*). Only seven genes (*ACTL7B*, *CCDC148*, *COX4I2*, *CRAT*, *IMMP2L, LIPC,* and *ZBTB7C*) were not found to be associated with tumor processes (Appendix A). Altogether, these observations indicate that genes recurrently targeted by viral insertion are a suitable source of candidate genes to be involved in HPV-associated oncogenesis.

## 4. Discussion

The observation that HPV insertion in integration hot spots may target cancer-related genes has been well documented [27,38,43,45]. We report here that the frequency of this event is observed in 5–10% of the non-recurrent integration sites dispersed throughout the genome and progressively increases proportionally with increasing HPV insertion recurrency levels. We show, in addition, that high-risk HPV16/18/45 more frequently target cancer-related genes than the other viral genotypes. Finally, we report that HPV18/45, exhibiting the highest frequency of targeting cancer-related genes, were rarely integrated at multiple chromosome sites in individual tumors. This correlates with the fact that HPV18/45 DNA is always found in the integrated form in carcinoma cells [36,46] and found in patients, on average, younger than those with other HPV-associated tumors [90]. These data suggest that the oncogenicity of HPV18/45 is stronger than that of other genotypes, for which a longer preinvasive step may allow the development of multiple integration events that are less critical for tumor progression. In accordance, Brant et al. [54] also observed HPV18 more frequently inserted into genes than HPV16. However, the nature of the targets does not seem to be different between the viral genotypes; although certain preferential hot spots for HPV18/45 were found at the *VMP1*, *MYC*, and *LRP1B* loci, no specific gene targets for these high-risk genotypes were identified.

Altogether, these data indicate that in addition to its impact leading to the constitutive expression of the E6/E7 viral oncoproteins, HPV integration acts in oncogenesis via direct interactions between the viral and host cell genomes. Our results provide significant features sustaining the dynamic process of HPV-driven carcinogenesis acting via the selection of driver integration clones that harbor HPV integrants at gene loci susceptible to tumor progression. Importantly, this constitutes a rational basis for the development of potential innovative therapies aiming to counteract the underlying processes via specific molecular interventions. We found that about 30% of the genes targeted by HPV insertion correspond to direct or indirect actionable targets.

Different processes may lead to viral DNA insertion into the genome and account for distinct integration patterns with various impacts on the HPV/host gene expression. There is growing evidence that HPV DNA integration takes place in part through microhomology-mediated repair (MHMR) mechanisms of DNA breaks [34,35,39,91,92,93,94], and this may account for the two main mechanisms of “direct” and “looping” integration processes identified thus far for viral DNA insertions [35,91,92]. HPV18/45 genotypes more frequently integrate following the “looping” or amplification processes, and this may account for the specificities observed for the integration pattern of these genotypes. Whether this is linked to the strong HPV18 E7 impact on cell proliferation [95] or has consequences on disease outcome remains to be determined. Besides genomic amplification, various mechanisms for gene overexpression through HPV DNA insertion have been described, such as viral enhancer insertions, inter-chromosomal translocation hijacking of host enhancers, and long-range cis-activation or chromatin remodeling [39,50,92,96]. Different processes leading to gene silencing have also been reported [33,34,39]. It is also mentioned that an indirect influence of HPV DNA inserts may favor the tumor process: for instance, in a large review of sequencing data from various types of cancers [47], an increase in mutations number within viral integration sites was observed, which might reflect chromatin features facilitating integration into regions with limited access to DNA repair or direct influence of integrated viral sequences on the proximal host genome.

The development of innovative therapeutic approaches using HPV as a target for molecular- [97] or immuno-therapy [98] has already provided encouraging results. It has also been suggested that host genes deregulated via HPV integration could be attractive targets for personalized therapy, such as the functional *ERBB2* and *RAD51* genes or *CD274*, *PDCD1LG2*, and *BRCA4*, which may provide a rational basis for immunotherapy [43].

The optimal characterization of the viral physical status in HPV-associated carcinomas constitutes information that can be clinically relevant. A negative outcome of HPV-associated tumors harboring integrated vs. exclusively episomal viral sequences has been observed [37,58,99]. More importantly, hybrid viral/host sequences represent a highly specific tumor marker, detectable in the blood [35,76,100,101], that may be useful in patient follow-up care [60]. Recently, the value of using circulating HPV DNA as a tumor marker predictive of tumor relapse following treatment of HPV-associated tumors has been clinically validated [102]. It is important to underline that NGS-based analyses provide, in a single experiment, a full characterization of the viral pattern, including the exact genotyping via the complete DNA sequence, the viral physical status, and the localization of the viral inserts into the host genome. The possibility of performing such analyses from a standard blood sample should facilitate its use in clinical practice.

## 5. Conclusions

In conclusion, the analysis of our database permitted us to show that (I) HPV18/45 DNA is rarely found at multiple sites in individual tumors, (II) recurrent insertions were not specific to HPV genotypes or tumor localizations, (III) the proportion of target genes implicated in oncogenesis increases proportionally with increasing HPV insertion recurrency levels and with the oncogenicity of the HPV genotypes, and (IV) 30% of the genes at HPV insertion loci correspond to potential therapeutic targets. The characterization of the full viral pattern in HPV-associated cancers provides data that can be used for diagnosis, follow-up, and personalized therapy. The systematic and prospective biobanking and informatic analyses of these data should help to improve the knowledge of HPV-associated carcinogenesis and favor the evolution of the traditional histological classification of HPV tumors toward a more relevant histo-molecular approach, as in other tumor types.

## Data Availability

All of the data collected and used for this study are provided in the Appendix A.

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
