# Peer review of "HPV DNA Integration at Actionable Cancer-Related Genes Loci in HPV-Associated Carcinomas"

_cancers, 2024, doi:10.3390/cancers16081584_

Round 1
Reviewer 1 Report
Comments and Suggestions for Authors
Dear authors, thank you for this interesting paper about HPV DNA integration site and prognostic significance.
Some points should be addressed
Please modify the citation management according to the journal style of reference
line 39: this is not true, please refer to IARC classification of high risk papillomavirus(carcinogenic type 1, 2A and 2B)
line 44-45: also refer to vaginal carcinoma and HPV, here as follows the most recent report about it: https://doi.org/10.1002/jmv.29474
line 50-51: DNA integration occurs in a frequent manner also in pre invasive lesions, please modify accordingly
liner 291: eliminate and modify the errors
very nice figures
The results are supported by references.
Thank you for your work and please amend the aforementioned points.
Comments on the Quality of English LanguageMinor
Author Response
Expert N° 1
We thank the expert for his positive evaluation of our work.
We think that we have amended our work according to the recommendations as detailed below. We have added the figure and the references requested. All of the modifications appear colored in the text.
The citation management has been modified
Line 29: The IARC classification of high-risk HPVs has been introduced
Line 36: The reference of HPV-associated vaginal cancer has been introduced.
Line 42: Data from literature do not support the statement that “DNA integration occurs in a frequent manner also in pre invasive lesions”. Only 5% of the HPV DNA integration cases collected in the present study correspond to pre invasive lesions. We have nevertheless modified the sentence “In preinvasive lesions, the viral genome generally replicates extra-chromosomally in the cell nucleus” for “commonly replicates extra-chromosomally”.
Lines 415-416. The status of TPRG1 has been modified as a “cell proliferation factor” and a new reference introduced in the table.
Reviewer 2 Report
Comments and Suggestions for Authors
General comments:
1. If this manuscript belongs to an article, what is the primary aim of this article?
2. If this is an original article, a simple summary does not seem necessary to be presented before the abstract.
3. What are the protocols and concise analysis in this article?
4. Do the results meet the research question reported in the abstract?
5. Can this study be understood from the abstract alone?
Other comments:
1. In the abstract, what’s the definition of “non-recurrent,” “weakly-recurrent,” and highly-recurrent.”?
2. In the introduction:
Can this huge database, which you have spent time and energy creating, meet or achieve the four purposes you want to establish? How many goals can the results of this analysis achieve?
3. In the materials and methods:
(A) Please supply a study flow diagram and list the inclusion and exclusion criteria more clearly.
(B) Please outline the statistical analysis process in your materials and methods.
(C) In Lines 157-158, you excluded the 157 genes with a very low expression value in the respective cell lines to avoid some false positives, but how can you avoid false negatives?
(D) In lines 158-161, please list the references about “DepMap.”
4. In the results:
What important findings can respond to the main purpose of this study?
5. In the conclusions:
Are the conclusions included supported by the results reported?
Comments on the Quality of English LanguageModerate editing of the English language required
Author Response
Expert N°2
We thank the expert for his positive evaluation of our work.
We think that we have amended our work according to the recommendations as detailed below. We have added the figure and the references requested. All of the modifications appear colored in the text.
- The primary aim of the article has been further explicated at the end of the introduction.
- The Simple Summary has been suppressed.
- The protocols and concise analysis have been further explicated in the beginning of the paragraph “Material and Methods”.
4-5) We think that the results meet the research question reported in the abstract. The main results of this study can be understood from the abstract alone
Other comments
1 In the abstract, the definitions of “non-recurrent,” “weakly-recurrent,” and highly-recurrent are provided
2 Introduction: we have more explicitly specified the main aim of the article (cf point 1) so that we think it is now easier to understand how the four purposes addressed by this study may be deduced from the analysis of the database.
3 Material and Methods
The paragraph Material and Methods has been largely modified. We have added a flow diagram (A) and listed more clearly the inclusion and exclusion criteria (B). We mainly used fisher tests for the statistical analysis, we detailed further in the methods.
(C) We rewrote to be clearer about this point: Depmap CRISPR score was considered in the analysis but not for screening. From the list of highly recurrent genes that had no reported cancer role on OncoKB, we evaluated the Depmap CRISPR score together with literature to look for new candidates associated to cancer.
(D) We have added the reference for the DepMap study
4 In the results, we have underlined the most important facts obtained from this analysis corresponding to the main aim of the study.
5 In the conclusion we have introduced a new paragraph indicating more explicitly how these conclusions are deduced from the results.
Round 2
Reviewer 1 Report
Comments and Suggestions for Authors
Dear authors
you have addressed all the questions reported.
now the paper deserves publication
Thank you
Author Response
Thank you for your positive evaluation.
Reviewer 2 Report
Comments and Suggestions for Authors
Comment to “HPV DNA integration at actionable cancer-related genes loci
In HPV-associated carcinomas”
Minor comment:
1. The study flow chart and table are crucial components of the research article, typically included in the main body and usually within the Methods section. They provide readers with a visual representation of the study design, participant recruitment process, and data collection procedures, and therefore, their accuracy and clarity are paramount.
2. In the Figure S1. Flowchart of database HPV integration sites analysis:
When drawing the study flow chart, some key points should be kept in mind:
a. Consider easy to read.
b. Specify case number.
c. Account for dropouts and losses to follow-up.
d. Review for accuracy.
e. Provide details on intervention.
f. Avoid overcrowding it with unnecessary details.
g. Ensure that all components are clearly labeled.
3. Please re-draw your study flow chart, paying particular attention to the Rt side blue boxes and the Lt side last box (bottom), as they seem to be causing confusion.
4. Additionally, in Table S1, please ensure that every space is filled with a "0" or "NA" to avoid any ambiguity. Explain every abbreviation under the table.
(for example: NA: Non-available). Add “all” at the bottom of the leftmost column.
5. Please revise reference 85, which you added as follows.
85. DepMap, Broad; Kocak, Mustafa (2023). Repurposing Public 23Q2. figshare. Dataset. https://doi.org/10.6084/m9.figshare.23600310.v3
Author Response
The flowchart has been updated and included in the section Material and Methods. The reference has also been updated.